# Structural insights into the mechanism and inhibition of transglutaminase-induced ubiquitination by the *Legionella* effector MavC

Yajuan Mu[1,4], Yue Wang [1,4], Yanfei Huang[1,4], Dong Li[1,4], Youyou Han[1], Min Chang[1], Jiaqi Fu[2], Yongchao Xie[1], Jie Ren[3], Hao Wang[1], Yi Zhang[1], Zhao-Qing Luo [2] & Yue Feng [1✉]

Protein ubiquitination is one of the most prevalent post-translational modifications, controlling virtually every process in eukaryotic cells. Recently, the *Legionella* effector MavC was found to mediate a unique ubiquitination through transglutamination, linking ubiquitin (Ub) to UBE2N through Ub[Gln40] in a process that can be inhibited by another *Legionella* effector, Lpg2149. Here, we report the structures of MavC/UBE2N/Ub ternary complex, MavC/UBE2N-Ub (product) binary complex, and MavC/Lpg2149 binary complex. During the ubiquitination, the loop containing the modification site K92 of UBE2N undergoes marked conformational change, and Lpg2149 inhibits this ubiquitination through competing with Ub to bind MavC. Moreover, we found that MavC itself also exhibits weak deubiquitinase activity towards this non-canonical ubiquitination. Together, our study not only provides insights into the mechanism and inhibition of this transglutaminase-induced ubiquitination by MavC, but also sheds light on the future studies into UBE2N inhibition by this modification and deubiquitinases of this unique ubiquitination.

[1] Beijing Advanced Innovation Center for Soft Matter Science and Engineering, Beijing Key Laboratory of Bioprocess, State Key Laboratory of Chemical Resource Engineering, College of Life Science and Technology, Beijing University of Chemical Technology, 100029 Beijing, China. [2] Purdue Institute for Inflammation, Immunology and Infectious Disease and Department of Biological Sciences, Purdue University, West Lafayette, IN, USA. [3] State Key Laboratory for Biology of Plant Diseases and Insect Pests/Key Laboratory of Control of Biological Hazard Factors (Plant Origin) for Agri-product Quality and Safety, Ministry of Agriculture, Institute of Plant Protection, Chinese Academy of Agricultural Sciences, 100081 Beijing, China. [4] These authors contributed equally: Yajuan Mu, Yue Wang, Yanfei Huang, Dong Li. ✉email: fengyue@mail.buct.edu.cn

Ubiquitination is one of the most widely used protein modifications in eukaryotic cells, regulating almost every essential cellular process[1]. Canonical ubiquitination is carried out by the actions of the E1, E2, and E3 enzymes, which function as a three-enzyme cascade to covalently attach ubiquitin (Ub) to a lysine residue of a protein substrate[2]. Prokaryotes do not contain the Ub system, however, co-option of the host Ub network is adopted by a variety of bacterial pathogens to support their own survivals[3]. This is usually executed by virulent factors, which behave as E3 Ub ligases[4], deubiquitinases (DUBs)[5,6], or enzymes that modify Ub or proteins involved in ubiquitination[7,8]. *Legionella pneumophila*, a Gram-negative pathogen that parasitizes free living protozoan hosts[9], is the causative agent of a severe, potentially fatal pneumonia known as Legionnaires' disease in humans[10,11]. The pathogen survives and replicates within host cells by creating a membrane-bound vacuole (the *Legionella*-containing vacuole, or LCV), the biogenesis of which requires the activity of over 330 *Legionella* substrates (effectors) translocated into host cells by the conserved Dot/Icm type IV secretion system[12–15].

Out of these effectors, many have been found to co-opt the host Ub network[14]. For example, LegU1, AnkB, LubX, and GobX all exhibit Ub E3 ligase activities[16–18]. SidC also defines a unique family of E3 ligases with a Cys-His-Asp catalytic triad[19]. Interestingly, recent studies showed that members of the SidE family effectors directly ubiquitylate several substrates in a unique two-step process without the need for E1 and E2 enzymes[20–22]. This non-canonical ubiquitination was accomplished through successive modifications of the R42 residue of Ub by the mono-ADP-ribosyltransferase (mART) and phosphodiesterase (PDE) domains of this family effectors[23]. Members of the SidE family effectors also contain a DUB domain, which is important for Ub dynamics on the LCV[6].

Recently, two *L. pneumophila* effectors MavC and MvcA, were identified as structural homologs of cycle inhibiting factor (Cif) effectors[24]. Cif from enteropathogenic *Escherichia coli* (EPEC) and Cif homolog in *Burkholderia pseudomallei* (CHBP) could induce mammalian cell growth arrest and actin stress fiber formation[7]. Members of the Cif family deamidate a conserved glutamine residue Q40 in Ub and the Ub-like (Ubl) protein NEDD8 (refs. [7,25]). Moreover, a separate study revealed that MavC, but not MvcA, is actually a transglutaminase that catalyzes covalent linkage of Ub to K92 and to a less extent, K94 of the E2 enzyme UBE2N via Q40 of Ub[26]. Thus, MavC could induce a mono-ubiquitination of UBE2N through transglutamination. Transglutaminases (TGs) are enzymes involved in protein cross-linking that catalyze a transamidation reaction between the γ-carboxamide group of a glutamine residue of one protein (the "acceptor" substrate) and an amine (the "donor" substrate), which can be either an ε-amino group of a lysine residue from another protein or a small molecule amine[27,28]. The reaction starts from the formation of a γ-glutamylthioester between the active site Cys residue of the TG and the Gln-containing "acceptor" substrate[28]. In the absence of an amine donor, the thioester could be hydrolyzed to produce the glutamate residue, which corresponds to a net deamidation reaction of the "acceptor" substrate. To our knowledge, this is the first report of transglutaminase activity of a Cif effector[26]. In addition, MavC differs from canonical Cif effectors in two other aspects. First, MavC only targets Ub as its substrate, in contrary to the exclusive preference for NEDD8 by canonical Cif effectors[7,29–31]. Second, Lpg2149, another *L. pneumophila* effector, directly inhibits the activity of MavC[24]. Although the structures of MavC, MvcA, and Lpg2149 have been solved[24], it remains elusive how this non-canonical ubiquitination or transglutamination is carried out by a Cif-like effector and how such activity is inhibited by Lpg2149.

Here, we report the structures of the MavC/UBE2N/Ub ternary complex, MavC/UBE2N–Ub (product) binary complex, and MavC/Lpg2149 binary complex, which provide important insights into the mechanism and inhibition of this transglutaminase-induced ubiquitination by MavC. Moreover, we found that MavC itself also exhibits weak activity to catalyze the reverse reaction, that is, the deubiquitination of this non-canonical ubiquitination product UBE2N–Ub. Taken together, this study reveals the molecular basis of this non-canonical ubiquitination and its inhibition by Lpg2149, and also provides a framework for future identification of enzymes which can catalyze and remove this unique ubiquitination, respectively.

## Results

**Overall structure of the MavC/UBE2N/Ub complex**. To understand the mechanism underlying MavC-catalyzed non-canonical ubiquitination, we cocrystallized MavC, UBE2N, and Ub, and solved the crystal structure of the MavC[C74A]/UBE2N/Ub ternary complex at a resolution of 2.93 Å (Fig. 1a–d, Supplementary Figs. 1a, b, 2a, 3a and Table 1). Following the nomenclature of the study of Valleau et al. [24], MavC comprises of a main domain, which is further divided into head and tail regions, and an insertion domain (residues 128–226) (Fig. 1a). The overall fold of MavC remains largely unchanged upon binding to UBE2N and Ub, with a root-mean-square deviation (RMSD) of 1.92 Å among 345 residues between MavC in the complex and the apo MavC (PDB code: 5TSC). The major conformational change occurs at their insertion domains, which show a rotation of around 34° with respect to their main domains between the two structures[32] (Supplementary Figs. 2a and 3b). This is a rigid-body movement of the insertion domain, because the insertion domains of the two structures could superimpose well with an RMSD of 0.64 Å among 99 residues (Supplementary Fig. 3c). The insertion domain of MavC contributes to interaction with UBE2N, which also interacts with the head region of the main domain of MavC simultaneously (Fig. 1a). Results from gel filtration assays indicated that the insertion domain, but not the main domain of MavC, could form a stable complex with UBE2N (Supplementary Fig. 4), suggesting that the insertion domain offers the major interacting surface for UBE2N. Structural alignments with available UBE2N structures reveal only slight conformational changes in the loop region containing the modification sites K92 and K94 (Supplementary Fig. 3e, f). However, in the structure, this loop region exhibits a high B-factor, indicating that this region tends to become disordered and undergo disorganization (Supplementary Fig. 3g). The structure of Ub in the MavC complex is also nearly identical to that of free Ub (Cα RMSD, 0.69 Å), except for the flexible C-terminal tail (Supplementary Fig. 3d). Different from other Cif effectors which bind Ub with only the head and tail regions, the insertion domain of MavC also interacts with Ub apart from the main domain (Fig. 1a, b). Moreover, the detailed interaction interfaces between MavC and Ub are different from those of other Cif effectors (to be described below).

**MavC binds Ub through four distinct patches**. The MavC–Ub binding buries 2580.7 Å² surface area, including four contacting regions (Fig. 1c, d). Although different Ub residues are engaged by MavC when compared with CHBP[30], we use similar nomenclature to name the four contacting areas of Ub as contact A1–C1 and the C-terminal contact (CTC). Contact A1 in Ub involves the hydrophobic patch formed by L8/T9/H68 (Fig. 1e), but I44, which is frequently used in protein–Ub binding, is not directly involved. The corresponding interface involves the N-terminal tail region of MavC, in which I31, L36, I43, and the aliphatic chain parts of N39 and E40 form hydrophobic interactions with

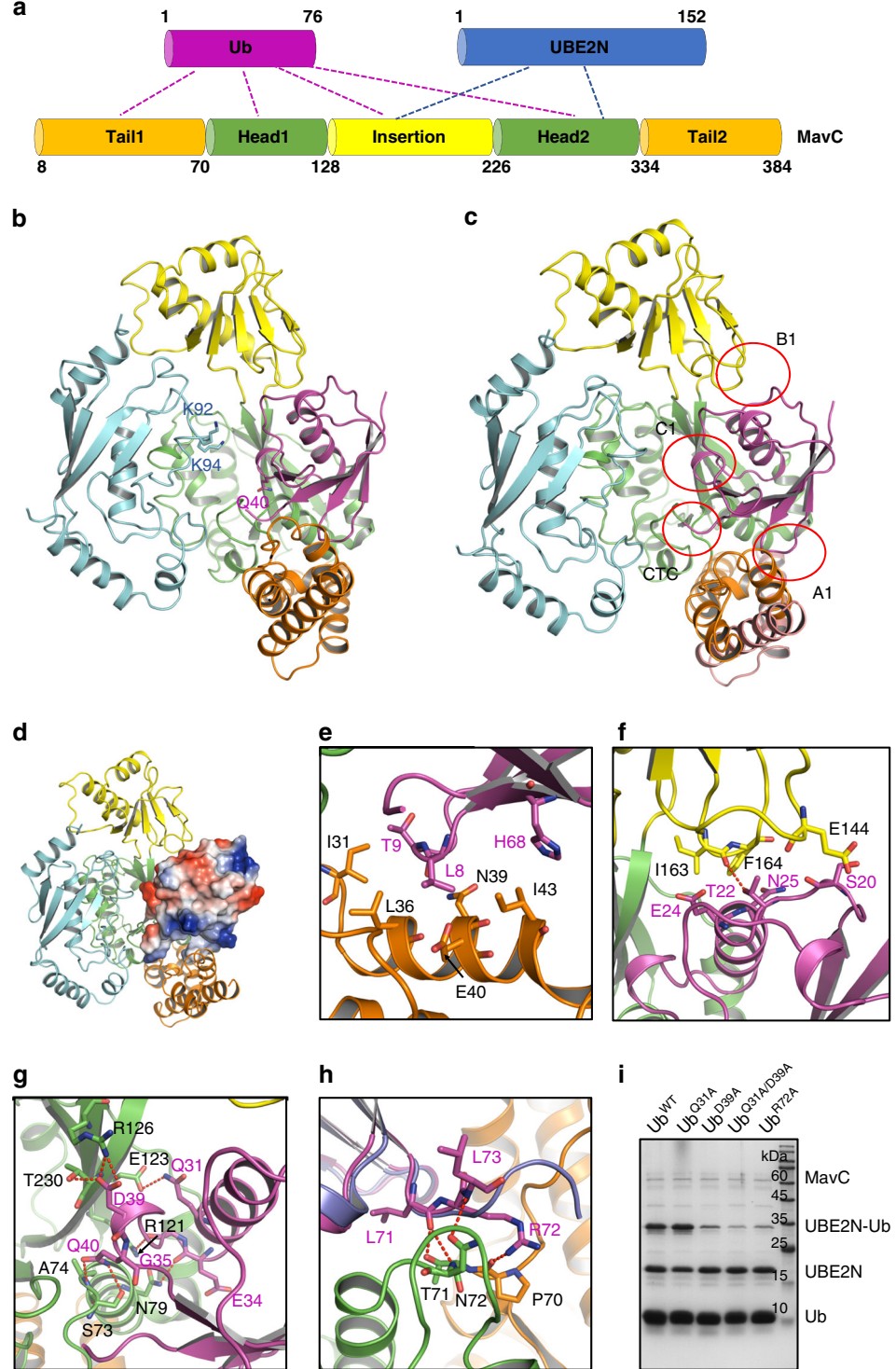

**Fig. 1 Overall structure of the MavC/UBE2N/Ub complex. a** Domain architecture of MavC showing interaction modes among MavC, UBE2N, and Ub.
**b** Overall structure of the MavC/UBE2N/Ub complex, colored as in **a**. Q40 of Ub, K92 and K94 of UBE2N are shown as sticks. **c** Overview of the interfaces
between MavC and Ub in the MavC/UBE2N/Ub complex. **d** Overall structure of the MavC/UBE2N/Ub complex, with Ub shown in the electrostatic surface
model. **e–g** Detailed interfaces of MavC–Ub interaction as marked in **c**, including A1 **e**, B1 **f** and C1 **g**. Hydrogen bonds are represented as red dashed lines.
**h** Detailed CTC interface of MavC–Ub interaction as marked in **c**. Ub in the CHBP/Ub structure is colored in blue, and aligned to that in the MavC/UBE2N/
Ub complex. **i** Mutations of the MavC-interacting residues in Ub decreased its ability in MavC-catalyzed ubiquitination. Ub and its mutants were incubated
with MavC and UBE2N for 1 h at 37 °C. Then the samples were subjected to Tricine gel, followed by Coomassie blue staining. Source data are provided as a
Source Data file. The experiment was repeated independently three times with similar results.

**Table 1 Data collection and refinement statistics.**

|  | MavC/UBE2N/Ub | MavC/UBE2N–Ub | MavC/Lpg2149 |
|---|---|---|---|
| *Data collection* |  |  |  |
| Space group | C222$_1$ | C2 | P2$_1$2$_1$2$_1$ |
| Cell dimensions |  |  |  |
| $a, b, c$ (Å) | 97.7, 105.9, 125.4 | 107.4, 97.2, 74.4 | 52.9, 83.8, 124.2 |
| $\alpha, \beta, \gamma$ (°) | 90.00, 90.00, 90.00 | 90.00, 103.27, 90.00 | 90.00, 90.00, 90.00 |
| Resolution (Å) | 50–2.92 (3.02–2.92)[a] | 50–2.39 (2.48–2.39) | 50–1.98 (2.05–1.98) |
| $R_{sym}$ or $R_{merge}$ | 0.098 (0.794) | 0.072 (0.325) | 0.108 (0.363) |
| $I/\sigma(I)$ | 26.6 (3.33) | 22.6 (5.54) | 25.9 (5.3) |
| Completeness (%) | 100.0 (100.0) | 100.0 (100.0) | 99.7 (97.4) |
| Redundancy | 12.0 (11.8) | 6.1 (6.0) | 10.6 (9.9) |
| *Refinement* |  |  |  |
| Resolution (Å) | 38.93–2.92 (3.03–2.92) | 30.94–2.39 (2.48–2.39) | 36.28–1.97 (2.05–1.97) |
| No. of reflections | 14,350 (1390) | 29,359 (2853) | 39,377 (3744) |
| $R_{work}/R_{free}$ | 0.2095/0.2617 | 0.1902/0.2448 | 0.1724/0.2058 |
| No. of atoms | 4753 | 5025 | 4366 |
| Protein | 4753 | 4816 | 3884 |
| Ligand/ion | 0 | 0 | 0 |
| Water | 0 | 209 | 482 |
| *B* factors | 53.62 | 43.62 | 32.22 |
| Protein | 53.62 | 43.79 | 31.57 |
| Ligand/ion |  |  |  |
| Water |  | 39.66 | 37.49 |
| R.m.s. deviations |  |  |  |
| Bond lengths (Å) | 0.003 | 0.010 | 0.009 |
| Bond angles (°) | 0.62 | 1.30 | 1.20 |

Values in parentheses are for highest-resolution shell.
[a]For each structure one crystal was used.

Ub. Contact B1 mainly involves the loop linking β2 (residues 12–16) and α1 (residues 23–34) of Ub and the loop spanning from L161 to K166 of the insertion domain of MavC (Fig. 1f). Contact B1 from Ub might contribute to the movement of the insertion domain of MavC in the complex structure. Contact C1 involves the α1 helix and the loop linking α1 to β3 (residues 42–44) of Ub, in which extensive hydrophilic interactions are involved (Fig. 1g). The deamidation site Q40 is also contained in this contact region, which may play an essential role in the presentation of Q40 to the catalytic center of MavC. Hydrogen bonds are formed between the sidechains of Ub$^{Q31}$ and MavC$^{E123}$, and between the sidechains of MavC$^{N79}$ and MavC$^{R121}$, and the backbone carbonyl oxygen atoms of Ub$^{E34}$ and Ub$^{G35}$, respectively (Fig. 1g). Moreover, the sidechain of D39 of Ub forms hydrophilic interactions with both the sidechains of R126 and T230 of MavC. The sidechain of Q40 also forms hydrogen bonds with the amide nitrogen atom of C74 (A74 here in the structure) and the sidechain hydroxyl of MavC$^{S73}$. The CTC also mainly contains hydrogen bond interactions (Fig. 1h). In the ternary complex structure, the density for the last three residues of Ub (R74-G76) are lacking, suggesting that they are not involved in the interaction with MavC.

Compared with the CHBP–Ub complex, the interaction between MavC and Ub displays several unique features. First, the relative positions of Ub to MavC/CHBP are different between the two structures (Supplementary Fig. 5a). Second, due to the different positions, many of the interactions between CHBP and Ub are not retained in the MavC/Ub complex[30]. Notably, the K11 surface in Ub, which features extensive hydrophilic interactions with CHBP, has been identified as most critical for CHBP interaction and deamidation[30] (Supplementary Fig. 5b). However, this surface does not form any hydrophilic interactions with MavC (Supplementary Fig. 5c). Third, Ub also interacts with the MavC-specific insertion domain. Last but not least, when the

Ubs in the two structures are superimposed, their C-terminal tails lie in different orientations relative to the two enzymes (Fig. 1h).

**Substrate specificity of MavC**. Known Cif/CHBP family members recognize exclusively, or show a preference for NEDD8 (refs. [30,31,33]), but MavC only deamidates Ub as revealed by the two studies[24,26]. To investigate the structural determinants for the substrate specificity, we analyzed the interaction details of MavC–Ub (Fig. 1e–h), and the structure and sequence alignment between Ub and NEDD8 (Supplementary Fig. 6a, b). Collectively, we hypothesized that Q31, D39, and R72 of Ub (E31, Q39, and A72 in NEDD8, respectively) might be important for Ub recognition by MavC. Then we tested the ubiquitination/deamidation activities of the Ub mutants of these residues. Ub mutants D39A and R72A both showed decreased ubiquitination (Fig. 1i) and deamidation (Supplementary Fig. 7a) activities. Although the Ub Q31A mutant itself did not show a markedly decreased activity, the Q31A/D39A double mutant exhibited lower activities than the D39A mutant both in the ubiquitination (Fig. 1i) and deamidation (Supplementary Fig. 7a) assays. These results suggest that the three Ub residues are pivotal for being recognized by MavC. Consistently, single or double mutations of the corresponding MavC residues E123 and R126, which interact with Ub$^{Q31}$ and Ub$^{D39}$, respectively, decreased both the deamidation (Supplementary Fig. 6c) and ubiquitination (Supplementary Fig. 6d) activities. Then we mutated E31, Q39, and A72 of NEDD8 to the corresponding residues of Ub to test whether these mutations could turn the non-targeted NEDD8 into a substrate of MavC. However, the NEDD8 mutant E31Q/Q39D/A72R was only a substrate with weak activity in both the deamidation (Supplementary Fig. 7b) and ubiquitination (Supplementary Fig. 7c) reactions catalyzed by MavC. Superimposition of the structure of NEDD8 with Ub in the MavC/UBE2N/Ub complex

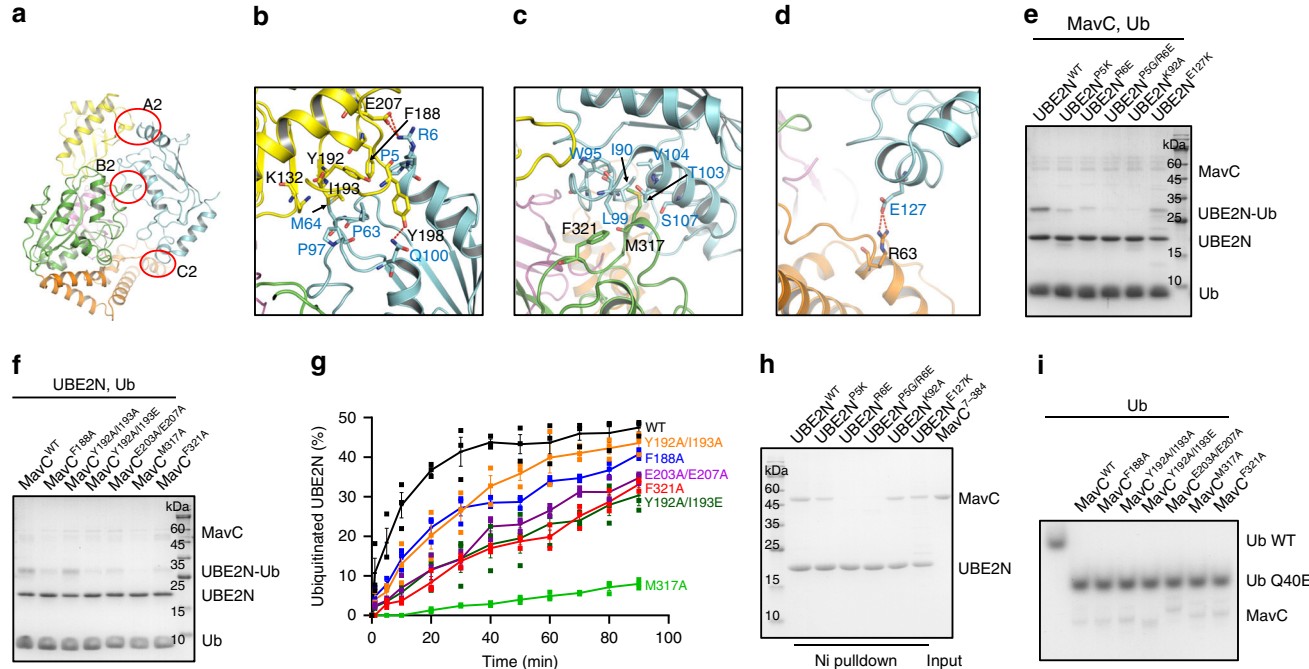

**Fig. 2 The interaction between MavC and UBE2N. a** Overview of the interface between MavC and UBE2N in the MavC/UBE2N/Ub complex. **b–d** Detailed interfaces of MavC–UBE2N interaction as marked in **a**, including A2 **b**, B2 **c**, and C2 **d**. Hydrogen bonds are represented as red dashed lines. **e** UBE2N and its mutants were incubated with MavC and Ub for 10 min at 37 °C. Then the samples were treated as in Fig. 1i. **f** Disrupting MavC–UBE2N interactions decreased the activity of MavC. MavC and the mutants were incubated with UBE2N and Ub for 10 min at 37 °C. Then the samples were treated as in Fig. 1i. **g** Kinetic analysis of MavC and the mutants targeting the UBE2N-binding surface. The percentage of ubiquitinated UBE2N was calculated by the amount of produced UBE2N–Ub divided by the sum of produced UBE2N–Ub and free UBE2N, at different reaction time points. Data shown are mean values ± SEM ($n = 3$ independent experiments). The representative gels used for the quantification are shown in Supplementary Fig. 8. **h** Ni pull-down experiment, combined with mutagenesis experiments to evaluate the roles of UBE2N residues in MavC–UBE2N interaction. **i** MavC mutants disrupting MavC–UBE2N interactions did not show dampened deamidase activity. MavC and its mutants were incubated with Ub for 2 h at 37 °C. Then the samples were subjected to native gel, followed by Coomassie blue staining. Source data are provided as a Source Data file. Experiments in **e**, **f**, **h**, and **i** were repeated independently three times with similar results.

showed that the sidechain of Ub$^{T22}$ forms a hydrogen bond with the carbonyl oxygen of MavC$^{I163}$, however, the corresponding residue in NEDD8 is K22, whose sidechain may cause steric and electrostatic clashes with the sidechain of MavC$^{K148}$ opposite to Ub$^{T22}$ in the structure (Supplementary Fig. 7d). Therefore, we mutated NEDD8$^{K22}$ to the corresponding Ub residue Thr in the background of the NEDD8 triple mutant and tested its ability as a substrate. The result showed that it could work as a substrate almost similar to Ub under our experimental conditions (Supplementary Fig. 7c). This NEDD8 mutant can be useful in future target discovery for MavC and other enzymes with similar functions[34].

**Mechanism of UBE2N recognition by MavC.** The MavC–UBE2N binding buries 2300.6 Å$^2$ surface area, including three contacting regions, namely contact A2–C2 (Fig. 2a). Contact A2, involving α1 (residues 6–18) and α2 (residues 100–113) helices, and the loop linking β3 (residues 51–57) and β4 (residues 68–71) of UBE2N, contains both hydrophobic and hydrophilic interactions. The sidechains of UBE2N$^{R6}$ and UBE2N$^{Q100}$ are hydrogen bonded to the sidechains of MavC$^{E207}$ and MavC$^{Y198}$ of the insertion domain, respectively. Moreover, hydrophobic interactions between P63/M64/P97 of UBE2N and Y192/I193 of MavC, and between UBE2N$^{P5}$ and MavC$^{F188}$ further anchor the complex conformation (Fig. 2b). Contact B2 involves α2 and the long loop (residues 72–99) linking β4 to α2 of UBE2N and the head domain of MavC. Hydrophobic interface on UBE2N, composed of I90, W95, T103, V104, and S107, surrounds M317 of MavC. L99 of UBE2N also forms hydrophobic interaction with F321 of MavC (Fig. 2c).

Contact C2 contains the electrostatic interaction between the sidechains of UBE2N$^{E127}$ and MavC$^{R63}$ of the N-terminal tail region (Fig. 2d).

MavC F188, Y192/I193, E203/E207, and the UBE2N P5/R6 mutations, designed to disrupt contact A2 interaction, all decreased the ubiquitination efficiency (Fig. 2e–g and Supplementary Fig. 8). Moreover, the activity of contact B2-deficient mutant of MavC (M317A) was decreased more severely (Fig. 2f, g). Another contact B2-deficient mutant of MavC (F321A) also showed a decreased activity. Contact C2 seems to play a minor role, as the UBE2N E127K mutation only slightly decreased the ubiquitination efficiency (Fig. 2e). Consistently, the above UBE2N mutants showed more or less defect in MavC binding (Fig. 2h). Notably, the UBE2N K92A mutant bound to MavC similarly as wildtype UBE2N. None of the above mutations interfered with the catalytic center of MavC because these mutants remained active in Ub deamidation (Fig. 2i).

**Conformational changes in UBE2N during the reaction.** Interestingly, in structure of the ternary complex, no marked conformational change was observed for the K92-containing loop region of UBE2N (Fig. 1b). However, B-factor analysis suggested that this region in the structure might become disordered and undergo disorganization (Supplementary Fig. 3g). That is, the conformational change of this region required to accomplish the ubiquitination reaction was not captured in the structure of the ternary complex. To gain insights into the catalytic process of this non-canonical ubiquitination, we purified the MavC-catalyzed non-canonical ubiquitination product UBE2N–Ub conjugate to

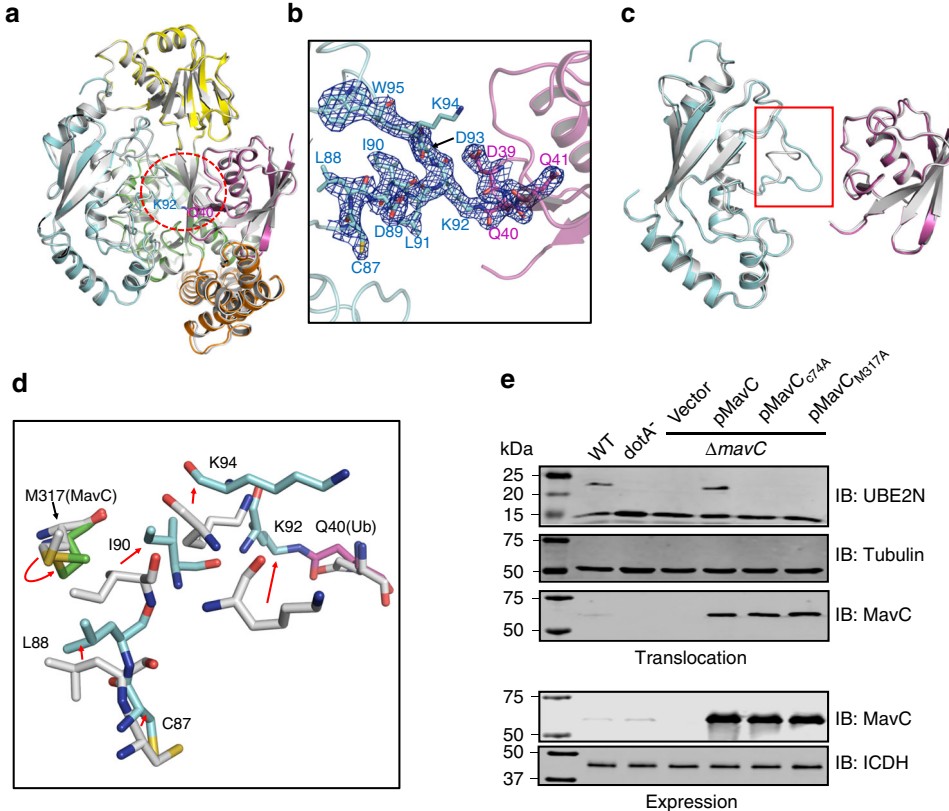

**Fig. 3 The conformational change of UBE2N during catalysis. a** Structural superimposition between the MavC/UBE2N–Ub binary complex and MavC/UBE2N/Ub ternary complex. MavC, UBE2N, and Ub in the MavC/UBE2N–Ub binary complex is colored as in Fig. 1a. The MavC/UBE2N/Ub ternary complex is colored in gray. UBE2N$^{K92}$ and Ub$^{Q40}$, which form a covalent bond in the MavC/UBE2N–Ub structure, are shown in sticks. The region of UBE2N which undergoes marked conformational change is indicated by a red circle. **b** Enlarged view of the UBE2N conformational change region in **a**. The 2Fo−Fc electron density map contoured at 1σ is shown in blue mesh for residues 87–95 of UBE2N and residues 39–41 of Ub in the MavC/UBE2N–Ub binary complex. **c** Detailed view of the conformational change of the K92/K94 loop of UBE2N. UBE2N and Ub molecules in the two structures are colored as in **a**. The region with the conformational change is highlighted in a red box. **d** Detailed view of the conformational change of UBE2N. The MavC/UBE2N–Ub binary complex and MavC/UBE2N/Ub ternary complex are colored as in **a**. Representative residues of the three proteins in the two structures are shown in sticks. The red arrows indicate the movement of the residues from the resting state to the catalytic state. **e** MavC$^{M317A}$ is defective in inducing UBE2N modification during *L. pneumophila* infection. Cells were infected with wild-type (WT) *L. pneumophila* or its *dotA* mutant defective in the Dot/Icm type IV transporter or *mavC* complementation strains. Saponin-soluble cell lysates were resolved by SDS–PAGE and probed with indicated antibodies to detect the ubiquitination of UBE2N (top) and the translocation of MavC and its mutants (middle). Bacterial lysates were probed for the expression of MavC and its mutants (bottom). Tubulin and ICDH were probed as loading control, respectively. Source data are provided as a Source Data file. The experiment was repeated independently three times with similar results.

homogeneity, and cocrystallized it with MavC$^{C74A}$. Then we solved the crystal structure of the MavC$^{C74A}$/UBE2N–Ub complex at a resolution of 2.39 Å (Fig. 3a and Supplementary Figs. 1c, d and 2b and Table 1). While the overall structures of the three proteins remain largely unchanged compared with the MavC/UBE2N/Ub complex (Fig. 3a), an isopeptide bond was clearly formed between the ε-amino group of UBE2N$^{K92}$ and Ub$^{Q40}$ in the MavC$^{C74A}$/UBE2N–Ub complex (Fig. 3b). Structural alignment with the MavC/UBE2N/Ub complex reveals a marked movement of K92/K94 loop region in the MavC$^{C74A}$/UBE2N–Ub structure (Fig. 3c), which forms a short helix in known structures of free UBE2N, to our knowledge. The extension of the K92/K94 loop towards the deamidated form of Ub$^{Q40}$ allows the reaction to be accomplished by MavC. Structural alignment also showed that M317 of MavC undergoes an obvious movement during the catalysis, which might play a key role in releasing the K92/K94 loop towards the active site (Fig. 3d). Consistently, kinetic analysis of the MavC mutants revealed that the hydrophobic and catalytically important interface around M317 plays the most important role in the ubiquitination (Fig. 2g), yet it is not required for Ub deamidation (Fig. 2i) activity of MavC. The hydrophobic interface

F188, Y192/I193, and F321 may play a minor role in its ubiquitination activity (Fig. 2g). Moreover, the charged interface E203/E207 also plays a minor role in the ubiquitination activity of MavC. We also examined the role of M317 in UBE2N ubiquitination during *L. pneumophila* infection. Whereas MavC expressed in the Δ*mavC* mutant induced UBE2N ubiquitination in infected cells, expression of MavC$^{M317A}$ in strain Δ*mavC* did not cause UBE2N modification in cells despite the mutant protein was properly expressed and translocated into host cells (Fig. 3e). Thus, UBE2N binding by MavC probably promotes the disorganization of the K92/K94 loop of UBE2N; the formation of the isopeptide bond with Ub completes the conformational changes of this loop.

**The mechanism of inhibition by Lpg2149.** Valleau et al. reported that Lpg2149 inhibits the activity of MavC by direct protein–protein interactions[24]. To investigate the inhibition mechanism of Lpg2149, we reconstituted the MavC–Lpg2149 complex in vitro and solved its crystal structure (Fig. 4a and Supplementary Figs. 1e, f and 2c and Table 1). The complex

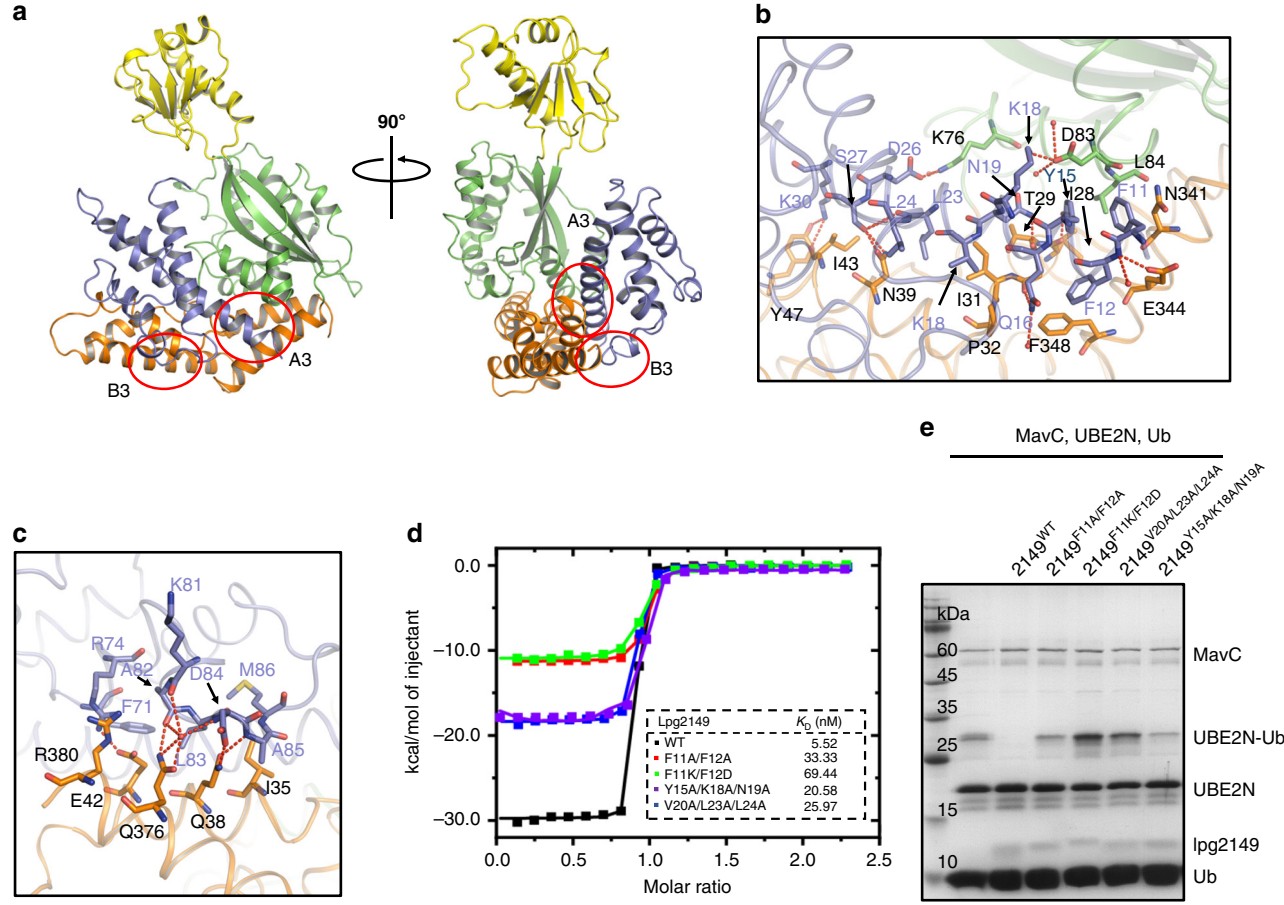

**Fig. 4 The inhibition mechanism of MavC by Lpg2149. a** Overall structure of MavC/Lpg2149. MavC is colored as in Fig. 1a and Lpg2149 is colored in blue. **b, c** Detailed interface of MavC–Lpg2149 interaction as marked in **a**, including A3 **b** and B3 **c**. **d** Isothermal titration calorimetry (ITC) assays to test binding of the Lpg2149 mutants to MavC. Representative binding curves and binding. affinities are shown. **e** Mutations that disrupt MavC–Lpg2149 interactions affect the inhibition capacity of Lpg2149. Lpg2149 and the mutants were incubated with MavC, UBE2N, and Ub for 2 h at 37 °C. Then the samples were treated as in Fig. 1i. Source data are provided as a Source Data file. The experiment was repeated independently three times with similar results.

structure revealed that MavC and Lpg2149 bind as a 1:1 dimer, consistent with the molecular weight of the complex measured by gel filtration coupled with static light scattering (SLS) analyses (Supplementary Fig. 9a). However, Lpg2149 crystallized as a domain-swapped dimer in the study of Valleau et al. [24] (Supplementary Fig. 9c). To verify its oligomerization state, we performed SLS analyses of Lpg2149 and MavC separately, which revealed that each of them exists as a monomer in solution (Supplementary Fig. 9a). Therefore, the dimer conformation of Lpg2149 in the previous study may be caused by crystal packing.

The overall fold of MavC remains unchanged upon binding to Lpg2149, with an RMSD of 2.03 Å among 323 residues between MavC in the complex and the apo MavC (PDB code: 5TSC). However, in the two structures, the insertion domains of MavC molecules display clearly different relative positions with respect to their main domains, further suggesting the flexibility of the tether linking the insertion and the main domain of MavC (Supplementary Fig. 9b). Lpg2149 has an overlapped binding site with Ub in MavC, but does not interfere with UBE2N binding (Supplementary Fig. 9d), indicating that Lpg2149 inhibits the activity of MavC through competing with Ub to bind MavC. Superimposition of the MavC molecules in the MavC/Lpg2149 and MavC/UBE2N–Ub complexes also shows a rotation of around 6° of the helical-bundle tail region (Supplementary Fig. 9d), where both Lpg2149 and Ub bind, indicating a binding-induced movement of the helical-bundle tail region.

Lpg2149 extensively interacts with MavC via multiple contact points, burying a surface area of 2229 Å$^2$. Lpg2149 interacts with both the head and tail region of the main domain of MavC mainly through two contacting regions, namely contact A3 and B3. Contact A3, involving the N-terminal long α1 helix (residues 11–38) of Lpg2149, contains six pairs of hydrogen bonds: the sidechain nitrogen of Lpg2149$^{Q16}$ with the main-chain carbonyl oxygen of MavC$^{G30}$, the amide nitrogen of Lpg2149$^{F12}$ with the sidechain of MavC$^{E344}$, the sidechain hydroxyl of Lpg2149$^{S27}$ with the sidechain of MavC$^{N39}$, the sidechain hydroxyl of Lpg2149$^{Y15}$ with the main-chain carbonyl oxygen of MavC$^{I28}$, and the sidechain of Lpg2149$^{N19}$ with the main-chain carbonyl oxygen of MavC$^{T29}$ (Fig. 4b). Electrostatic interactions are formed between the sidechains of Lpg2149$^{D26}$ and MavC$^{K76}$, and between the sidechains of Lpg2149$^{K18}$ and MavC$^{D83}$. Moreover, hydrophobic interactions between F11/F12 of Lpg2149 and N341/F348 of MavC, and between V20/L23/L24 of Lpg2149 and I31/I35/N39 of MavC further anchor the complex conformation (Fig. 4b). In contact B3 which involves the α3/α4 helix region (residues 70–77/79–82) of Lpg2149, the sidechain of MavC$^{Q38}$ forms two hydrogen bonds with the sidechain oxygen of Lpg2149$^{D84}$ and the amide nitrogen of Lpg2149$^{A85}$ (Fig. 4c). The sidechain of MavC$^{Q376}$ interacts with the main-chain carbonyl oxygen of Lpg2149$^{K81}$ and the amide nitrogen of Lpg2149$^{D84}$ through two water-mediated hydrogen bonds. The sidechains of Lpg2149$^{R74}$ and MavC$^{E42}$ interact with electrostatic

interactions, and hydrophobic interactions are formed between F71/L83/M86 of Lpg2149 and the aliphatic chain parts of I35/E34/Q38 of MavC (Fig. 4c).

Lpg2149 F11A/F12A, F11K/F12D, Y15A/K18A/N19A, and V20A/L23A/L24A mutations, designed to disrupt MavC–Lpg2149 interaction, all showed decreased MavC binding as detected both by isothermal titration calorimetry (ITC) assay (Fig. 4d and Supplementary Fig. 10) and IC50 analysis (Supplementary Fig. 11). Ubiquitination (Fig. 4e) and deamidation (Supplementary Fig. 12a) activity assays also indicated that these Lpg2149 mutants all show decreased inhibition capacities. Consistently, the ubiquitination activities of the corresponding mutants of MavC, which are designed to disrupt MavC–Lpg2149 binding, remained unchanged in the presence of Lpg2149 (Supplementary Fig. 12b).

**Deubiquitination activity of MavC.** During the co-crystallization of MavC and UBE2N–Ub conjugate, interestingly, we found that wildtype MavC, but not the C74A mutant, can actually cleave the UBE2N–Ub conjugate back to UBE2N and Ub (Fig. 5a). That is, MavC itself could reverse this modification on UBE2N. Consistently, MavC$^{C74A}$ exhibited a relatively high binding affinity of 5.91 μM to UBE2N–Ub by the ITC assay (Fig. 5c). Due to the binding specificity of MavC towards UBE2N–Ub, it was not surprised to find that MavC did not exhibit DUB activity towards canonical K48- and K63-diubiquitin (Fig. 5a). Then we examined both the transglutaminase and DUB activities of MavC through a series of reactions in which UBE2N or UBE2N–Ub and MavC were added at different molar ratios (Fig. 5b, d). The results showed that both the transglutaminase and DUB activities started to be detected when the molar ratios between MavC and substrate (UBE2N and UBE2N–Ub, respectively) were higher than 1:640 under our experimental conditions (Fig. 5b, d). Moreover, we performed kinetic analysis of MavC under the same experimental conditions as in Fig. 2g, but extended the reaction time up to 3 h (Fig. 5e). The results showed that the quantity of UBE2N–Ub product first increases from 0 to around 120 min, and then slightly decreases probably due to the DUB activity of MavC. Therefore, transglutaminase-induced ubiquitination, but not the reverse reaction, is the dominant activity of MavC. Thus, MavC also exhibits weak DUB activity for the ubiquitination reaction catalyzed by itself. As this activity is also dependent on the active site Cys74 (Fig. 5a) and based on the catalytic mechanism of MavC, this reverse process should also first involve the γ-glutamylthioester intermediate formed by MavC$^{C74}$ and Q40 of Ub, which was then hydrolyzed to a free glutamate residue (also corresponds to a net deamidation of Ub) (Fig. 5f).

## Discussion

MavC represents the first Cif effector to exhibit transglutaminase activity, and therefore, it also represents a family of TGs, which is structurally different from all known TGs of both eukaryotic and bacterial origins[35] (Supplementary Fig. 13). MavC catalyzes the transglutamination reaction between Ub and UBE2N, leading to UBE2N ubiquitination in an unconventional form. Importantly, UBE2N has been suggested as the specific substrate of MavC[26]. Sequence alignment between UBE2N and its homologs, in which C87 and K92 of UBE2N are also conserved, revealed that several of the essential MavC-interacting residues of UBE2N are not conserved in other structurally similar E2 enzymes, explaining its specific recognition by MavC (Supplementary Fig. 14).

Humans have about 40 E2 enzymes which are involved in the transfer of Ub or Ubl proteins[36]. Apart from functioning as a carrier of Ub during the ubiquitination cascade, E2s perform a variety of other important functional roles, such as directly engaging a target protein or regulating the activities of other

enzymes[37,38]. Moreover, the activities of E2s are regulated by various mechanisms, including transcriptional/translational control, non-covalent interactions by other proteins, and covalent post-translational modifications[36]. UBE2N forms heterodimers with UBE2V1 or UBE2V2, in combination with a variety of E3 ligases, to catalyze the elongation of K63-type Ub chains, which are important for various signaling pathways[39]. Non-canonical ubiquitination of UBE2N by MavC abolishes its activity in the formation of K63-type polyubiquitin chains, which dampens NF-κB signaling in the initial phase of *L. pneumophila* infection[24,26]. Despite extensive efforts, we were not successful in crystallizing the UBE2N–Ub conjugate, probably due to the induced local disorganization of the K92/K94 loop. However, we have obtained insights into the mechanism of inhibition of UBE2N activity by this unique ubiquitination from the MavC/UBE2N–Ub structure. Superimposition between the structures of UBE2N–Ub conjugate in our study and the canonically activated form of UBE2N bound to Ub indicated that this unique ubiquitination at K92 of UBE2N causes a steric clash to restrain Ub loading through its G76 to the active site C87 of UBE2N (Supplementary Fig. 15), thus explaining the inhibition of E2 activity of UBE2N by this modification.

Of note is that MavC itself also exhibits weak activity to cleave the UBE2N–Ub conjugate when higher amounts of protein were included in the reactions. Given the high level similarity between MavC and MvcA[24], which was very recently found to be a DUB specific for UBE2N–Ub[40], the DUB activity of MavC is not completely unexpected. Since the amount of MavC translocated into host cells is extremely low and the cellular concentrations of Ub is considered to be higher than that of the UBE2N–Ub conjugate produced by MavC in *L. pneumophila*-infected cells, the DUB activity of MavC may not be physiologically significant during bacterial infection. Previous studies have shown that MvcA does not catalyze the transglutamination between UBE2N and Ub[24,26]. That is, MavC primarily catalyzes the non-canonical ubiquitination with also weak activity in the opposite reaction, but MvcA is an obligate DUB for this ubiquitination. Structural comparisons between MavC/UBE2N–Ub and MvcA/UBE2N–Ub[40] reveal that while the main domain of MavC/MvcA and Ub portions superimpose well between the two structures when MavC and MvcA are aligned, the insertion domain of MavC/MvcA and UBE2N molecules display markedly distinct orientations between the two structures (Supplementary Fig. 16a). This further supports the view that the surface of Ub bound by MavC and MvcA is highly conserved[24], and the active site residues and the residues involved in Ub binding are also relatively conserved between MavC and MvcA[40]. However, the residues involved in UBE2N binding are largely different between MavC and MvcA, especially in their insertion domains (Supplementary Figs. 16b, c and 17a). In the meantime, the UBE2N residues engaged by the insertion domains of MavC and MvcA also do not overlap with each other except for R6. Therefore, we propose that the insertion domain not only endows the non-canonical UBE2N ubiquitination/deubiquitination activity of MavC/MvcA, but also distinguishes the ubiquitination activity of MavC and the deubiquitination activity of MvcA. Interestingly, the insertion domain also displays the lowest residue identities among the domains of the two proteins (Supplementary Fig. 17b). However, it still awaits further investigation how two structural homologs could exhibit the opposite activities.

UBE2N ubiquitination by MavC abolishes its activity, which dampens NF-κB signaling and likely other cellular processes regulated by K63-type polyUb chains. In macrophages, NF-κB activation induced by PAMPs is detrimental to bacterial colonization, but for *L. pneumophila*, NF-κB activity appears to benefit

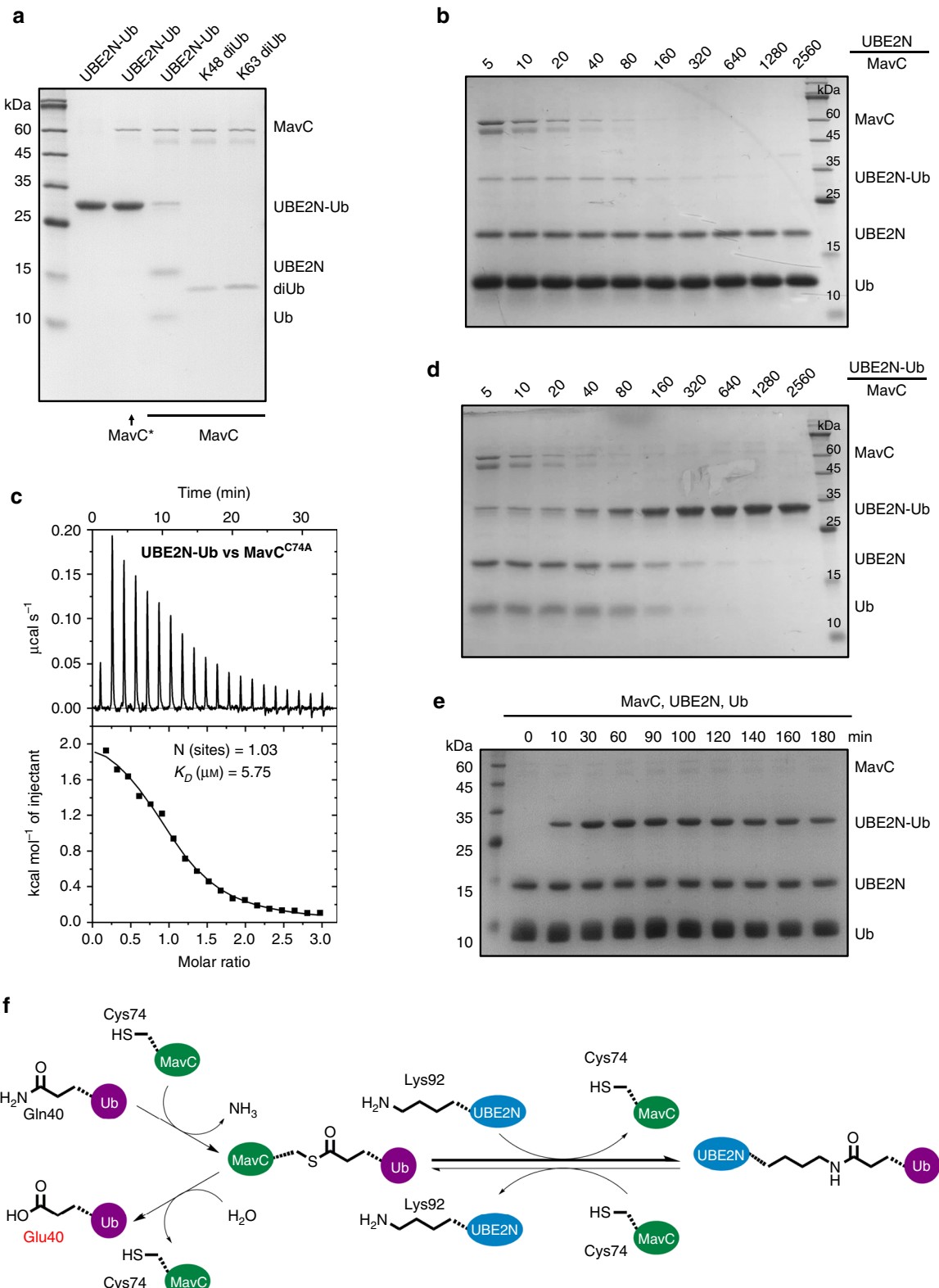

the bacterium in multiple aspects[41]. The *mavC* gene is present only in some isolates of *L. pneumophila*[42,43]. Furthermore, the *mvcA* gene is present in all strains that harbor *mavC* and these two genes are syntenic in all cases. The *MavC* gene is induced in the exponential phase and continued to the post-exponential phase, that is, MavC primarily function in the initial phase of the infection[24,26]. The regulation of MavC imposed by MvcA, the so-called meta-effector[44,45], is achieved by the induction of the *MvcA* gene in later phases of infection[40]. Therefore, MavC and MvcA temporally regulate UBE2N activity during *L. pneumophila* infection by the differential expression of these two genes at different stages of its intracellular life cycle. Moreover, the regulation of MavC and MvcA is further complicated by Lpg2149, which inhibits the activity of both enzymes by direct binding[24]. It

**Fig. 5 MavC also exhibits deubiquitinase activity towards UBE2N–Ub. a** Purified UBE2N–Ub made by MavC, K48, and K63 diubiquitin were incubated with MavC or MavC$^{C74A}$. MavC* in the figure indicates MavC$^{C74A}$. **b** A series of in vitro reactions containing UBE2N, Ub, and MavC at the indicated molar ratios were set up and allowed to proceed for 1 h at 37 °C. **c** MavC interacts with UBE2N–Ub with a $K_D$ of 5.91 μM. A representative binding curve by ITC assay to test binding of UBE2N to MavC$^{C74A}$ is shown with the binding affinity. **d** A series of in vitro reactions containing UBE2N–Ub conjugate and MavC at the indicated molar ratios were set up and allowed to proceed for 1 h at 37 °C. **e** Kinetic analysis of MavC-catalyzed modification of UBE2N. MavC was incubated with UBE2N and Ub at 37 °C for the indicated amounts of time. **f** In the transglutamination reaction, the nucleophilic MavC$^{C74}$ attacks Ub$^{Q40}$, to form a thioester intermediate, which further reacts with the amine donor from UBE2N$^{K92}$ (mainly) to form an intermolecular isopeptide bond between UBE2N and Ub. In the reverse reaction, the isopeptide bond within the UBE2N–Ub conjugate is attacked by MavC$^{C74}$, resulting in the release of UBE2N and the formation of the MavC–Ub thioester intermediate, which could further react with a water molecule to give the deamidated form of Ub$^{Q40}$. The transglutamination activity is the dominant activity of MavC. Source data are provided as a Source Data file. Experiments in **a**, **b**, **d**, and **e** were repeated independently three times with similar results.

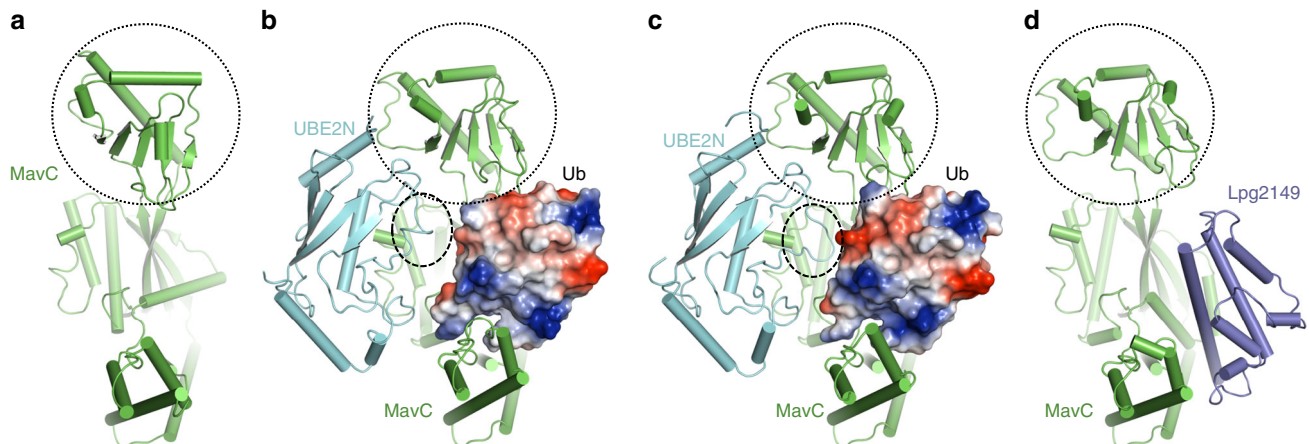

**Fig. 6 Overview of the structures in this study. a–d** Overall structures of apo MavC (**a**, PDB: 5TSC), MavC/UBE2N/Ub complex **b**, MavC/UBE2N-Ub complex **c**, and MavC/Lpg2149 complex **d**. Ub is shown in electrostatic surface model. The insertion domain of MavC and the K92 loop region of UBE2N are marked with circles.

has been reported that in bacteriological medium, the Lpg2149 expression only becomes detectable in the early exponential phase, suggesting that it functions when the bacteria began to replicate[40]. Because proteins being translocated by the Dot/Icm system are in their unfolded forms[46], Lpg2149 may inhibit the activity of MavC and MvcA by preventing their translocation through forming protein complexes in bacterial cells or by blocking the activity of translocated proteins in host cells. Yet, the exact time and the extent of its inhibition of MavC and MvcA still needs further investigation. For the temporal regulation of MavC by MvcA, the reversal of UBE2N ubiquitination by MvcA could be detected at 6 h post-infection[26,40], indicating that the inhibition of UBE2N is relieved during later phases of the intracellular life cycle of the bacteria. Taken together, these results suggest that the MavC–MvcA–Lpg2149 system is sophisticatedly regulated to condition the cellular environment to best accommodate intracellular replication of *L. pneumophila*. The mechanisms of the inhibition of UBE2N activity by transglutaminase-induced ubiquitination by MavC and of its further inhibition by Lpg2149 revealed by our results (Fig. 6) will serve as a foundation for future identification of enzymes that catalyze this unique ubiquitination.

## Methods

**Protein expression and purification**. Sequences of all the primers used in this study are listed in Supplementary Table 1. The full-length and various segments of *L. pneumophila* MavC were amplified by PCR and cloned into pGEX6p-1 vector to produce GST-tagged fusion proteins with a PreScission Protease cleavage site between GST and the target proteins. The MavC mutants were generated by two-step PCR and were subcloned, overexpressed, and purified in the same way as wild-type protein. Particularly, the MavC clone with deletion of residues 128–226 was

made by bridging PCR, and a "GSG" sequence was added between the two MavC fragments. The proteins were expressed in *Escherichia coli* (*E. coli*) strain BL21 and induced by 0.2 mM isopropyl-β-ᴅ-thiogalactopyranoside (IPTG) when the cell density reached an OD$_{600\,nm}$ of 0.8. After growth at 16 °C for 12 h, the cells were harvested, re-suspended in lysis buffer (1 × PBS, 2 mM DTT, and 1 mM PMSF) and lysed by sonication. The cell lysate was centrifuged at 17,940 × g for 45 min at 4 °C to remove cell debris. The supernatant was applied onto a self-packaged GST-affinity column (2 mL glutathione Sepharose 4B; GE Healthcare) and contaminant proteins were removed with wash buffer (1 × PBS and 2 mM DTT). The fusion protein was then digested with PreScission protease at 18 °C for 2 h. The protein with an additional five-amino-acid tag (GPLGS) at the N-terminus was eluted with buffer containing 25 mM HEPES pH 7.5, 200 mM NaCl, and 2 mM DTT. The eluant was concentrated using an Ultrafree 5000 molecular-weight cutoff filter unit (Millipore) and further purified using a Superdex-200 (GE Healthcare) column equilibrated with a buffer containing 10 mM Tris–HCl pH 8.0, 200 mM NaCl, and 5 mM DTT. The purified protein was analyzed by SDS–PAGE. The fractions containing the target protein were pooled and concentrated.

The fragment of human UBE2N cDNA (residues 1–152) was cloned into pGEX6p-1 and pET28a vectors to produce GST-tagged fusion protein, or His-tagged fusion protein, respectively. The fusion protein was induced in *E. coli* Rosetta (DE3) similarly as MavC. Recombinant His-tagged protein was purified by Ni-affinity column chromatography and gel filtration chromatography (Superdex-75 column) in buffer containing 10 mM Tris–HCl pH 8.0, 200 mM NaCl, 5 mM DTT. The UBE2N mutants were generated by two-step PCR and were subcloned, overexpressed, and purified in the same way as wild-type UBE2N.

Wildtype and mutants of Ub and NEDD8 used in this study were cloned into pGEX6p-1 vectors or pET28a vectors, to produce GST-tagged fusion proteins or His-tagged fusion proteins, respectively. The proteins were purified according to the above protocols for GST-tagged or His-tagged proteins, concentrated and stored in −80 °C until use. Wildtype and mutants of Lpg2149 were cloned into pGEX6p-1 vectors to produce GST-tagged fusion proteins and purified according to the protocols for GST-tagged proteins.

**Preparation of the MavC-catalyzed UBE2N-Ub product**. The UBE2N–Ub product was produced in a 20 mL MavC-catalyzed reaction containing 0.08 mg MavC, 3.2 mg UBE2N, and 8 mg His-tagged Ub at 37 °C for 2 h. The buffer is 50 mM Tris

pH 7.5 and 50 mM NaCl. After the reaction, the reaction mix was subjected to Ni-column followed by anion exchange and gel filtration chromatography in a buffer containing 10 mM Tris–HCl pH 8.0, 200 mM NaCl, and 5 mM DTT.

**Crystallization, data collection, and structure determination**. For the MavC/UBE2N/Ub ternary complex, MavC[7–384] (C74A), UBE2N, and Ub were incubated at a molar ratio of 1:1:2, in which the final concentration of MavC[7–384] (C74A) was 28 mg/mL. After overnight incubation, crystals of MavC/UBE2N/Ub complex were grown at 18 °C by mixing an equal volume of the complex protein with reservoir solution containing 0.1 M lithium chloride, 20% w/v polyethylene glycol (PEG) 3350, pH 6.8. The prepared MavC-catalyzed UBE2N–Ub product was mixed with MavC[7–384] (C74A) at a molar ratio of 1:1, in which the final concentration of MavC[7–384] (C74A) was 24 mg/mL. After overnight incubation, crystals of MavC/UBE2N–Ub complex were grown at 18 °C by mixing an equal volume of the protein complex with reservoir solution containing 0.1 M KCl, 0.02 M Tris pH 7.0, and 20% w/v PEG 4000. The MavC[7–384] (C74A) and Lpg2149[11–114] were concentrated in 10 mM Tris–HCl pH 8.0, 200 mM NaCl, and 5 mM DTT, and mixed at a molar ratio of 1:1, in which the final concentration of MavC[7–384] (C74A) was also 24 mg/mL. After overnight incubation, crystals of MavC/Lpg2149 complex were grown at 18 °C by mixing an equal volume of the protein with reservoir solution containing 0.2 M Ammonium nitrate, 20% w/v PEG3,350, pH 6.2. The crystals appeared overnight and grew to full size in about one week. The crystals were cryoprotected in the reservoir solution containing 15% glycerol before its transferring to liquid nitrogen.

All the data were collected at SSRF beamlines BL17U1 and BL19U1 (ref. [47]) with a wavelength of 0.979 Å, integrated and scaled using the HKL2000 package[48]. Further processing was carried out using programs from the CCP4 suite[49]. All the three structures were solved by molecular replacement with the structure of MavC (PDB: 5TSC), UBE2N (PDB: 1J7D), Lpg2149 (PDB: 5DPO), and Ub (PDB: 1UBQ) as templates. The structures were refined with several rounds of COOT[50] and PHENIX[51]. Final Ramachandran statistics: 95.76% favored, 3.73% allowed, and 0.51% outliers for MavC/UBE2N/Ub ternary complex; 98.33% favored, 1.5% allowed, and 0.17% outliers for MavC/UBE2N–Ub binary complex; 97.9% favored, 1.89% allowed, and 0.21% outliers for MavC/Lpg2149 complex. Data collection and structure refinement statistics are summarized in Table 1. All of the structural illustrations were generated using the software PyMOL.

**ITC binding assay**. The dissociation constants of binding reactions of Lpg2149 or Lpg2149 mutants with MavC were determined by ITC using a MicroCal ITC200 calorimeter. Proteins were desalted into the working buffer (20 mM HEPES pH 7.5, 200 mM NaCl). The titration was carried out with 19 successive injections of 2 µL Lpg2149 at the 0.2 mM concentration, spaced 120 s apart, into the sample cell containing the MavC (residues 7–384) at the 0.02 mM concentration at 25 °C. The Origin software was used for baseline correction, integration, and curve fitting to a single site binding model. The dissociation constant of binding reaction of UBE2N–Ub with MavC[C74A] (residues 7–384) was determined similarly.

**Multi-angle light scattering (MALS)**. Protein (100 µL) at 1 mg/mL was injected into a Superdex-75 column (GE Healthcare) equilibrated with the running buffer containing 10 mM Tris–HCl pH 8.0, 200 mM NaCl, 5 mM DTT. The chromatography system was coupled to an 18-angle light scattering detector (Wyatt Technology) for data collection. Data were collected every 0.5 s at a flow rate of 0.5 mL/min. Data analysis used program ASTRA 6.1. The figure was drawn using Origin 8.0.

**In vitro UBE2N ubiquitination assay**. All the ubiquitination reactions were carried out in a 25 µL reaction system and in buffer containing 50 mM Tris pH 7.5 and 50 mM NaCl. To test the activities of the MavC and Ub mutants proposed to disrupt MavC–Ub interaction, 0.4 µg MavC or its mutants, 4 µg UBE2N and 10 µg Ub or its mutants were incubated at 37 °C for 1 h. To test the substrate specificity, the reaction system was the same as above except that NEDD8 or its mutants instead of Ub were added in the mix in several reactions and the incubation condition is 37 °C and 2 h for all the reactions. To test the activities of the MavC and UBE2N mutants proposed to disrupt MavC–UBE2N interaction, the reaction system was the same as above but incubated at 37 °C for 10 min. To test the activities of the MavC and Lpg2149 mutants proposed to disrupt MavC–Lpg2149 interaction, 0.6 µg MavC or its mutants, 4 µg UBE2N, 10 µg Ub, and 0.4 µg Lpg2149[11–114] or its mutants were incubated at 37 °C for 2 h. After the reaction, all the samples were separated by Tricine gel.

**Kinetics analysis of MavC and its mutants**. The kinetics analysis of MavC and its mutants were carried out in a 25 µL reaction system and in the above buffer. 0.2 µg MavC or its mutants, 6 µg UBE2N, and 10 µg Ub were incubated at 37 °C for the indicated time, respectively. Then the samples were subjected to tricine gel and visualized by Coomassie blue staining. ImageJ was used to quantify the amount of UBE2N–Ub and the remaining UBE2N for each sample. The percentage of ubiquitinated UBE2N, that is, the ratio of produced UBE2N–Ub to the sum of free UBE2N and produced UBE2N–Ub at each time point was calculated. The final figure was produced by the Graphpad Prism 8.0.1 software.

**IC50 assay for the Lpg2149 mutants**. The IC50 assays for the Lpg2149 and its mutants were carried out in a 25 µL reaction system and in buffer containing 50 mM Tris pH 7.5 and 50 mM NaCl. In the reaction, 0.3 µg MavC and 6 µg UBE2N were first incubated with various concentrations of Lpg2149 or its mutants obtained by serial dilutions at 37 °C for 15 min, and then 10 µg Ub was added into the reaction system and further incubated for another 15 min. The reaction samples were subjected to tricine gel and visualized by Coomassie blue staining. ImageJ was used to quantify the amount of UBE2N–Ub for each sample. The amount of UBE2N–Ub generated by the reaction with Lpg2149 of indicated concentrations was then normalized to that generated by MavC with no Lpg2149 on each gel. The data were fitted and IC50 values were calculated by the GraphPad Prism 8.0.1 according to the dose–response model with variable slope.

**In vitro deamination assay**. All the deamination reactions were carried out in a 30 µL reaction system and in buffer containing 50 mM Tris pH 7.5 and 50 mM NaCl. All the reactions except those containing Lpg2149 were incubated at 37 °C for 2 h. To test the activities of the MavC mutants proposed to disrupt MavC–UBE2N interaction, 2 µg MavC or its mutants and 10 µg Ub were added. To test the activities of the MavC mutants proposed to disrupt MavC–Ub interaction, 3 µg MavC or its mutants and 10 µg Ub were added. To test the activities of the Ub mutants proposed to disrupt MavC–Ub interaction, 4 µg MavC and 10 µg Ub or its mutants were added. To test the substrate specificity, 6 µg MavC and 20 µg His-tagged Ub/NEDD8 or its mutants were added. For the reactions testing the inhibitory effect of Lpg2149, Lpg2149 WT, and it mutants were incubated with MavC and Ub or not, for 3 h at 37 °C. All the reaction mixtures were immediately separated on a 10% native-PAGE gel (pH 8.8 and pH 10.0 for reactions containing MavC/Lpg2149 mutants and His-tagged Ub/NEDD8 mutants, respectively) in an ice-water bath.

**In vitro Ni-column pull-down assay**. To detect the binding between MavC and UBE2N, 15 µL Ni beads were equilibrated with 200 µL buffer (50 mM Tris, 50 mM NaCl, pH 7.5) by two times. Then 50 µg His-tagged UBE2N[1–152] or its mutants and no-tagged WT MavC[7–384] were mixed with the beads at 4 °C for 1 h. After incubation, the beads were washed four times with the buffer containing 50 mM Tris pH 8.0, 300 mM NaCl, 30 mM imidazole. And then, 35 µL SDS–PAGE loading buffer was added into the beads and the samples were analyzed by Tricine gel. The experiment was repeated three times.

**Gel filtration-binding assay**. The MavC[128–226] or MavC[7–127/227–384] and UBE2N purified as described above were subjected to gel filtration analysis (Superdex-200, GE Healthcare). They were mixed with a molar ratio of about 1:1 and incubated at 4 °C overnight before the gel filtration analysis in buffer containing 10 mM Tris–HCl pH 8.0, 200 mM NaCl, and 5 mM DTT. Samples from relevant fractions were applied to SDS–PAGE and visualized by Coomassie blue staining.

**Deubiquitination assays**. For the deubiquitination activity of MavC, 0.8 µM WT or C74A mutant of MavC[1–482], 9.2 µM UBE2N–Ub, or K63/K48 di-Ub were incubated at 37 °C for 10 min in the same buffer as the ubiquitination assay. K48 and K63 di-Ub molecules (UC-200B and UC-300B, respectively) were obtained from Boston Biochem. The transglutaminase and DUB activities of MavC were also assayed through a series of reactions in which UBE2N (fixed at 6 µM) or UBE2N–Ub (fixed at 6 µM) and MavC were added at different molar ratios at 37 °C for 1 h.

**UBE2N ubiquitination by MavC during *L. pneumophila* infection**. *Legionella* strains used in this study were derivatives of Philadelphia 1 strain Lp02 and were grown and maintained on charcoal-yeast extract (CYE) plates or in ACES-buffered yeast extract (AYE) broth as described earlier[52]. Genes were inserted into pZL507 (ref. [53]) for complementation experiments. Raw264.7 cells and U937 cells were purchased from ATCC were cultured in RPMI1640 medium supplemented with 10% FBS.

For infection experiments, all *L. pneumophila* strains were grown overnight in AYE broth to postexponential phase judged by the optical density of the cultures (OD[600 nm] = 3.2–3.8) and by increase in bacterial motility, then 0.2 mM IPTG was added into bacterial cultures to induce the expression of MavC and its mutants on pZL507 at 37 °C for 3 h before infection. Raw264.7 cells or differentiated U937 macrophages were infected with an multiplicity of infection (MOI) of 15 at 37 °C for 2 h. Infected cells washed three times with PBS were lysed with 0.2% saponin for 30 min on ice. Saponin-soluble fractions were probed with UBE2N and MavC-specific antibodies for the modified UBE2N and the translocated MavC, respectively.

*Antibodies*: Anti-UBE2N (Thermo Fisher Scientific, cat. no. 37-1100), 1:1000; anti-MavC[26], 1:5000; anti-tubulin (Developmental Studies Hybridoma Bank, E7) 1:10,000; anti-ICDH[53], 1:10,000.

**Reporting summary**. Further information on research design is available in the Nature Research Reporting Summary linked to this article.

## Data availability

Coordinates and structure factors for the complexes have been deposited in the Protein Data Bank (PDB) under accessions: 6KFP, MavC/UBE2N/Ub; 6KG6, MavC/UBE2N-Ub;

and 6K3B, MavC/Lpg2149. Structures of MavC (PDB: 5TSC), UBE2N (PDB: 1J7D), Lpg2149 (PDB: 5DPO), NEDD8 (1NDD), and Ub (PDB: 1UBQ) were referenced in the manuscript. The source data underlying Figs. 1i, 2e–i, 3e, 4e, 5a, b and d, e and Supplementary Figs. 4a, b, 6c, d, 7a–c, 8a–g, 11a–f and 12a, b are provided as a Source Data file. Other data are available from the corresponding author upon reasonable request.

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

## Acknowledgements

We thank Jianwei Zeng for his help in structure refinement, and Maojun Yang and Lei Liu for the discussions about the mechanisms of MavC. We would like to thank the staff at beamlines BL17U1 and BL19U1 of the Shanghai Synchrotron Radiation Facility for their assistance with data collection. We would like to thank the Tsinghua University Branch of China National Center for Protein Sciences Beijing and Shilong Fan for providing facility support for X-ray diffraction of the crystal samples. This work was supported by the National Natural Science Foundation of China (31822012), the National key research and development program of China (2017YFA0506500), Beijing Nova program, the

Fundamental Research Funds for the Central Universities (XK1802-8) and the National Institutes of Health grants R01AI127465 and R01GM126296.

## Author contributions

Y.F. conceived, designed, and supervised the project. Y.M., Y.W., Y.H. (Yanfei Huang), D.L., Y.H. (Youyou Han), and M.C. purified the proteins, grew, and optimized the crystals. Y.F., Y.H. (Yanfei Huang), Y.W., and H.W. collected the diffraction data. Y.F. and Y.Z. solved the structures. Y.M., Y.W., Y.H. (Yanfei Huang), D.L., Y.H. (Youyou Han), M.C., Y.X., and J.R. performed all the in vitro activity analysis and binding assays. J.F. performed UBE2N ubiquitination assays during bacterial infection. Y.H. (Youyou Han) and M.C. contributed equally to this work. Y.F. analyzed the data and wrote the original manuscript. Y.F. and Z.L. revised the paper with assistance from all the authors.

## Competing interests

The authors declare no competing interests.
