## [Peer Review File · Nature Communications]

Reviewers' comments:

Reviewer #4 (Remarks to the Author):

Review

Structural insights into the mechanism and inhibition of transglutaminase-induced ubiquitination catalyzed by the Legionella effector MavC by Mu et al.

The MavC protein catalyzes the non-canonical mono-ubiquitination of the ubiquitin (Ub) conjugating E2 enzyme Ube2N in a two-step reaction: First the Ub Q40 side chain is deamidated and subsequently an isopeptide linkage of the resulting Ub E40 is formed with the K92 side chain in Ube2N. This reaction is inhibited by the effector protein Lpg2149.

The manuscript by Mu et al. describes crystal structures of three different MavC states: MavC in complex with Ube2N and Ub representing the substrate binding state, MavC bound to a Ube2N-Ub isopeptide representing the product state, and MavC associated with Lpg2149 representing an inhibited state. These structures together with deamination and ubiquitination assays as well as kinetic and biophysical analyses allowed the authors to identify residues critical for substrate, product and inhibitor binding and to elucidate the mechanisms of MavC activity and inhibition.

Although I was not part of the first round of the review process, I feel that the authors have largely addressed the reviewer's comments in a satisfactory manner.

However, two co-authors of the present manuscript (Jiaqi Fu and Zhao-Qing Luo) have recently published in EMBO J the crystal structure of the highly homologous (at least 50% sequence identity) Legionella protein Mvca in complex with the Ube2N-Ub adduct, with Zhao-Qing Luo even being a corresponding author of the EMBO J paper.

Although Mvca is shown to catalyze preferentially the deubiquitination reaction and thus seems to differ in its catalytic activity from MavC, the published structure is virtually identical to the MavC:Ube2N-Ub complex described in this manuscript. In particular, the key conformational changes, i.e. rotation of the insertion domain in MavC and unfolding of the K92 loop in Ube2N, identified in this manuscript are also present in the Mvca:Ube2N-Ub complex. However, this is not discussed in the present manuscript. Moreover, this questions how significant the present structure is in light of the already published structure.

Major comments:

1. The authors should therefore present structural comparisons, properly cite and discuss the published structure and describe the significance of their findings and elaborate on the advances they made considering the already published structure.
2. Line 245: MavCK92 should be UBE2NK92 !
3. The order of Figures 2 and 3 is confusing in the text. Fig. 3e and g are mentioned before 3f and 3a-d.
4. Line 248 has to be modified since the short helix in Ube2N is not present in the Mvca complex either (s. above).
5. Line 262: Ube2N binding by MavC alone is obviously not sufficient to induce conformational changes of the K92 loop, since these changes are only visible in the MavC:Ube2N-Ub, but not in

the ternary MavC:Ube2N:Ub complex.

6. It would improve clarity to color all MavC structure with the same color code as in Fig. 1a,b.

7. The contrast of almost all gels and blots should be improved.

Minor comments:

1. Line 118/119: The authors should state the angle of rotation between the insertion domains in both structures.

2. Throughout the paper, the authors refer to secondary structure elements (e.g. line 148). However, in none of the structure representations the secondary structure elements are labeled. It would be useful to use residue numbers in the text or labeled secondary structure elements in the figures.

3. The authors should make clear which figure displays deamination and which ubiquitination activity (e.g. line 185 should read "ubiquitination (Fig. 1j) and deamidation (Fig. S7a) activities" etc.).

4. Line 186f. : The deamination activities of Ub Q31A, D39A and Q31A/D39A (Fig. S7a) seem similar, but not their ubiquitination activities (Fig. 1j). This should be discussed, and the authors should specify which activity they refer to.

5. Line 235: The authors should explain in more detail what they mean by "does not represent a reacting conformation"

6. Line 249: Would that be Q40 or E40, i.e. before or after transamination?

7. Line 254: The authors should emphasize that M317 is crucial for ubiquitination but not for transamination.

8. Line 287: The conformational change seems to be quite small. As above it would be helpful to state an angle of rotation here.

9. Lines 405/406 are unclear to me.

10. Fig. 1: It would be nice to enlarge Fig. 1b and e. Maybe one of the views in Fig. 1c and Fig. 1d could be moved to the Supplement.

11. Fig. S7a,b: Labeling deamidated species with an asterisk would considerably improve the clarity of these figures.

12. It would be nice to have the same color code in Fig. 4d and S11a.

Reviewers' comments:

Reviewer #4 (Remarks to the Author):

Review

Structural insights into the mechanism and inhibition of transglutaminase-induced ubiquitination catalyzed by the Legionella effector MavC by Mu et al.

The MavC protein catalyzes the non-canonical mono-ubiquitination of the ubiquitin (Ub) conjugating E2 enzyme Ube2N in a two-step reaction: First the Ub Q40 side chain is deamidated and subsequently an isopeptide linkage of the resulting Ub E40 is formed with the K92 side chain in Ube2N. This reaction is inhibited by the effector protein Lpg2149.

The manuscript by Mu et al. describes crystal structures of three different MavC states: MavC in complex with Ube2N and Ub representing the substrate binding state, MavC bound to a Ube2N-Ub isopeptide representing the product state, and MavC associated with Lpg2149 representing an inhibited state. These structures together with deamination and ubiquitination assays as well as kinetic and biophysical analyses allowed the authors to identify residues critical for substrate, product and inhibitor binding and to elucidate the mechanisms of MavC activity and inhibition.

Although I was not part of the first round of the review process, I feel that the authors have largely addressed the reviewer's comments in a satisfactory manner.

However, two co-authors of the present manuscript (Jiaqi Fu and Zhao-Qing Luo) have recently published in EMBO J the crystal structure of the highly homologous (at least 50% sequence identity) Legionella protein MvcA in complex with the Ube2N-Ub adduct, with Zhao-Qing Luo even being a corresponding author of the EMBO J paper.

Although MvcA is shown to catalyze preferentially the deubiquitination reaction and thus seems to differ in its catalytic activity from MavC, the published structure is virtually identical to the MavC:Ube2N-Ub complex described in this manuscript. In particular, the key conformational changes, i.e. rotation of the insertion domain in MavC and unfolding of the K92 loop in Ube2N, identified in this manuscript are also present in the MvcA:Ube2N-Ub complex. However, this is not discussed in the present manuscript. Moreover, this questions how significant the present structure is in light of the already published structure.

We appreciate the reviewer's high evaluation of our work and constructive suggestions. We have addressed all of the comments listed below and have revised the manuscript in accordance with the reviewer's comments, which have significantly improved our manuscript.

We agree with the reviewer that the structure of MvcA complex published in the EMBO J paper is similar to the MavC/UBE2N-Ub structure in our study and this similarity should be further discussed in our manuscript. As the reviewer pointed out, although structurally similar, MavC and MvcA show opposite biochemical activities. The structural study in the EMBO J paper has detailed the mechanism of the reversal reaction by MvcA, while our results

provide important insights into the forward reaction (formation of UBE2N-Ub) by MavC. Moreover, as mentioned by the reviewer, our study has also made new discoveries not known from the published studies, which we summarized as follows. First, our work has identified the residues within the insertion domain of MavC critical for binding to the substrate UBE2N, which differ greatly from those of MvcA (Please see below). Second, we identified the essential role of MavC^{M317} in catalysis. Third, our results revealed that Lpg2149 inhibits the activity of MavC by competing with Ub for MavC binding. This mechanism is also applicable for the inhibition of MvcA by Lpg2149 due to the conservation between MavC and MvcA. Finally, we found that MavC itself has deubiquitinase activity *in vitro*, which suggests that MavC and MvcA evolved from a common ancestor and diverged to carry out reactions with opposite biological consequences.

In the revised manuscript, we have further compared and discussed our MavC/UBE2N-Ub structure with the MvcA/UBE2N-Ub complex described in the EMBO J paper and have highlighted the significance of our findings. Please see our point-by-point responses below.

Major comments:

1. The authors should therefore present structural comparisons, properly cite and discuss the published structure and describe the significance of their findings and elaborate on the advances they made considering the already published structure.

Our responses:

The reviewer raised a very good point. In the Discussion section, we have presented structural comparisons, discussed the published structure and described the significance of our findings and the advances made by our study in the revised manuscript.

“Structural comparisons between MavC/UBE2N-Ub and MvcA/UBE2N-Ub¹ reveal that while the main domain of MavC/MvcA and Ub portions superimpose well between the two structures when MavC and MvcA are aligned, the insertion domain of MavC/MvcA and UBE2N molecules display markedly distinct orientations between the two structures (Supplementary Fig. 16a). This further supports the view that the surface of Ub bound by MavC and MvcA is highly conserved², and the active site residues and the residues involved in Ub binding are also relatively conserved between MavC and MvcA¹. However, the residues involved in UBE2N binding are largely different between MavC and MvcA, especially in their insertion domains (Supplementary Fig. 16b-c and 17a). In the meantime, the UBE2N residues engaged by the insertion domains of MavC and MvcA also do not overlap with each other except for R6. Therefore, we propose that the insertion domain not only endows the non-canonical UBE2N ubiquitination/deubiquitination activity of MavC/MvcA, but also distinguishes the ubiquitination activity of MavC and the deubiquitination activity of MvcA. Interestingly, the insertion domain also displays the lowest residue identities among the domains of the two proteins (Supplementary Fig. 17b). However, it still awaits further investigation how two structural homologs could exhibit the opposite activities.”

2. Line 245: MavCK92 should be UBE2NK92 !

Our responses:

We are terribly sorry for this mistake. We have modified it and also carefully read through the manuscript to correct any errors.

3. The order of Figures 2 and 3 is confusing in the text. Fig. 3e and g are mentioned before 3f and 3a-d.

Our responses:

Following the reviewer's suggestion, we have changed the arrangement of Figures 2 and 3 to make the order of these two figures correct. The original Fig. 3e, 3g, 2f, 3f, and 3h have been changed to Fig. 2f, 2g, 2h, 2i, and 3e, respectively.

4. Line 248 has to be modified since the short helix in Ube2N is not present in the MavC complex either (s. above).

Our responses:

We apologize for our negligence of the MavC complex and have modified this sentence as, "which forms a short helix in known structures of free UBE2N, to our knowledge." in the revised manuscript.

5. Line 262: Ube2N binding by MavC alone is obviously not sufficient to induce conformational changes of the K92 loop, since these changes are only visible in the MavC:Ube2N-Ub, but not in the ternary MavC:Ube2N:Ub complex.

Our responses:

The reviewer raised a very good point. Although the conformational changes of the K92 loop was not visible in the ternary complex, B-factor analysis showed that this loop region exhibits a high B-factor value (the original Fig. S3f and now Fig. S3g), indicating that this region tends to become disordered and undergo disorganization. Therefore, we propose that UBE2N binding by MavC alone is not sufficient to induce the conformational changes of the K92 loop observed in the MavC/UBE2N-Ub complex, but probably promotes the disorganization of this region. Subsequent formation of the isopeptide bond with Ub further completes the conformational changes of this loop. We have modified the sentence as above in the revised manuscript.

6. It would improve clarity to color all MavC structure with the same color code as in Fig. 1a,b.

Our responses:

The reviewer raised a very good point. We have used the same code in all MavC structural figures as that in Fig. 1a, b.

7. The contrast of almost all gels and blots should be improved.

Our responses:

We have improved the contrast of all the gels and blots in the revised manuscript.

Minor comments:

1. Line 118/119: The authors should state the angle of rotation between the insertion domains in both structures.

Our responses:

The angle of rotation between the insertion domains in both structures is around 34°, which have been included into the revised manuscript.

2. Throughout the paper, the authors refer to secondary structure elements (e.g. line 148). However, in none of the structure representations the secondary structure elements are labeled. It would be useful to use residue numbers in the text or labeled secondary structure elements in the figures.

Our responses:

Following the reviewer's suggestions, we have added residue numbers where secondary structure elements are mentioned in the revised manuscript.

3. The authors should make clear which figure displays deamination and which ubiquitination activity (e.g. line 185 should read “ubiquitination (Fig. 1j) and deamidation (Fig. S7a) activities” etc.).

Our responses:

Following the reviewer's suggestion, we have made clear about the activities the figures are displaying throughout the manuscript.

4. Line 186f. : The deamination activities of Ub Q31A, D39A and Q31A/D39A (Fig. S7a) seem similar, but not their ubiquitination activities (Fig. 1j). This should be discussed, and the authors should specify which activity they refer to.

Our responses:

We apologize for the poor clarity of the original Fig. S7a. Following the reviewer's suggestion below, we have labeled the deamidated species with asterisks in Fig. S7a, from which we can see that the deamidation activity of the three mutants followed the order of Q31A>D39A>Q31A/D39A. This is consistent with their ubiquitination activity. Following the reviewer's suggestion, we have clearly described the activity the figures intended to demonstrate.

5. Line 235: The authors should explain in more detail what they mean by “does not represent a reacting conformation”

Our responses:

Following the reviewer's suggestion, we have explained this description in more detail in the revised manuscript and changed the sentence to "Interestingly, in structure of the ternary complex, no marked conformational change was observed for the K92-containing loop region of UBE2N (Fig. 1b). However, B-factor analysis suggested that this region in the structure might become disordered and undergo disorganization (Supplementary Fig. 3g). That is, the conformational change of the K92 loop region required to accomplish the ubiquitination was not captured in the structure of the ternary complex."

6. Line 249: Would that be Q40 or E40, i.e. before or after transamination?

Our responses:

The reviewer raised a very good point. It would be the deamidated form of Ub^{Q40} here. Regarding that it is referring to a residue of Ub, we have changed "Q40" to "deamidated form of Ub^{Q40}" in the revised manuscript.

7. Line 254: The authors should emphasize that M317 is crucial for ubiquitination but not for transamination.

Our responses:

We have emphasized that M317 is crucial for ubiquitination but not for ubiquitin deamidation, and cited the figure panels with the new order.

8. Line 287: The conformational change seems to be quite small. As above it would be helpful to state an angle of rotation here.

Our responses:

We have stated the angle of rotation in the revised manuscript.

9. Lines 405/406 are unclear to me.

Our responses:

We apologize for the ambiguity of these sentences and have clarified them in the revised manuscript (now lines 422-430).

"The *MavC* gene is induced in the exponential phase and continued to the post-exponential phase, that is, MavC primarily function in the initial phase of the infection^{2,3}. The regulation of MavC imposed by *MvcA*, the so-called meta-effector^{4,5}, is achieved by the induction of the *MvcA* gene in later phases of infection¹. Therefore, MavC and *MvcA* temporally regulate UBE2N activity during *L. pneumophila* infection by the differential expression of these two genes at different stages of its intracellular life cycle. Moreover, the regulation of MavC and *MvcA* is further complicated by Lpg2149, which inhibits the activity of both enzymes by direct binding²."

10. Fig. 1: It would be nice to enlarge Fig. 1b and e. Maybe one of the views in Fig. 1c and Fig. 1d could be moved to the Supplement.

Our responses:

The reviewer raised a very good point. We have enlarged the original Fig. 1b and e, and moved one of the views in Fig. 1c and Fig. 1d to the Supplement as Fig. S3a and S6b, respectively.

11. Fig. S7a,b: Labeling deamidated species with an asterisk would considerably improve the clarity of these figures.

Our responses:

The reviewer raised a very good point. We have labeled the deamidated species with asterisks in these two panels in the revised manuscript.

12. It would be nice to have the same color code in Fig. 4d and S11a.

Our responses:

The reviewer raised a very good point. We have used the same color code as Fig. 4d in Fig. S11a.

Again, we wish to thank the reviewer for her/his constructive comments and suggestions, which have helped us to improve our manuscript greatly.

References

1. Gan, N. et al. Legionella pneumophila regulates the activity of UBE2N by deamidase-mediated deubiquitination. *EMBO J*, e102806 (2019).
2. Valleau, D. et al. Discovery of Ubiquitin Deamidases in the Pathogenic Arsenal of Legionella pneumophila. *Cell Rep* **23**, 568-583 (2018).
3. Gan, N., Nakayasu, E.S., Hollenbeck, P.J. & Luo, Z.Q. Legionella pneumophila inhibits immune signalling via MavC-mediated transglutaminase-induced ubiquitination of UBE2N. *Nat Microbiol* **4**, 134-143 (2019).
4. Kubori, T., Shinzawa, N., Kanuka, H. & Nagai, H. Legionella metaeffector exploits host proteasome to temporally regulate cognate effector. *PLoS Pathog* **6**, e1001216 (2010).
5. Urbanus, M.L. et al. Diverse mechanisms of metaeffector activity in an intracellular bacterial pathogen, Legionella pneumophila. *Mol Syst Biol* **12**, 893 (2016).